# Anti-inflammatory and Anti-oxidant Activity of Hidrox^®^ in Rotenone-Induced Parkinson’s Disease in Mice

**DOI:** 10.3390/antiox9090824

**Published:** 2020-09-03

**Authors:** Rosalba Siracusa, Maria Scuto, Roberta Fusco, Angela Trovato, Maria Laura Ontario, Roberto Crea, Rosanna Di Paola, Salvatore Cuzzocrea, Vittorio Calabrese

**Affiliations:** 1Department of Chemical, Biological, Pharmaceutical and Environmental Sciences, University of Messina, Viale Ferdinando Stagno D’Alcontres, 31, 98166 Messina, Italy; rsiracusa@unime.it (R.S.); rfusco@unime.it (R.F.); salvator@unime.it (S.C.); 2Department of Biomedical and Biotechnological Sciences, University of Catania, Via S. Sofia, 89, 95123 Catania, Italy; mary-amir@hotmail.it (M.S.); marialaura.ontario@ontariosrl.it (M.L.O.); calabres@unime.it (V.C.); 3Oliphenol LLC., 26225 Eden Landing Road, Unit C, Hayward, CA 94545, USA; robertocrea48@gmail.com; 4Department of Pharmacological and Physiological Science, Saint Louis University School of Medicine, Saint Louis, MO 63104, USA

**Keywords:** polyphenols, hydroxytyrosol, neurodegenerative disease, neuroinflammation

## Abstract

Background: In developed countries, the extension of human life is increasingly accompanied by a progressive increase in neurodegenerative diseases, most of which do not yet have effective therapy but only symptomatic treatments. In recent years, plant polyphenols have aroused considerable interest in the scientific community. The mechanisms currently hypothesized for the pathogenesis of Parkinson’s disease (PD) are neuroinflammation, oxidative stress and apoptosis. Hydroxytyrosol (HT), the main component of Hidrox^®^ (HD), has been shown to have some of the highest free radical evacuation and anti-inflammatory activities. Here we wanted to study the role of HD on the neurobiological and behavioral alterations induced by rotenone. Methods: A study was conducted in which mice received HD (10 mg/kg, i.p.) concomitantly with rotenone (5 mg/kg, o.s.) for 28 days. Results: Locomotor activity, catalepsy, histological damage and several characteristic markers of the PD, such as the dopamine transporter (DAT) content, tyrosine hydroxylase (TH) and accumulation of α-synuclein, have been evaluated. Moreover, we observed the effects of HD on oxidative stress, neuroinflammation, apoptosis and inflammasomes. Taken together, the results obtained highlight HD’s ability to reduce the loss of dopaminergic neurons and the damage associated with it by counteracting the three main mechanisms of PD pathogenesis. Conclusion: HD is subject to fewer regulations than traditional drugs to improve patients’ brain health and could represent a promising nutraceutical choice to prevent PD.

## 1. Introduction

Parkinson’s disease (PD) is a neurodegenerative disease triggered by environmental and genetic factors. The degeneration of dopaminergic neurons in the area of the substantia nigra (SN) and the reduction of dopamine levels in the striatum area represent the main characteristics of PD that lead to symptoms such as tremor at rest, inelasticity, postural alteration and bradykinesia [1]. Moreover, patients with this pathology have formations known as Lewy bodies (LB), consisting mainly of cytoplasmic aggregations of α-synuclein [2,3]. 

Over the past two decades, inflammation in PD has received great interest by many researchers, highlighting microglial activation, cytokine production and oxidative damage in the in vivo model and post-mortem [4,5]. Therefore, neuroinflammation represents a key role in the progressive degeneration of dopaminergic neurons and in the pathogenesis of PD [6,7]. This is demonstrated by the discovery of high levels of some proinflammatory cytokines, such as interleukin (IL)-1β, tumor necrosis factor (TNF)-α, and many others, in the samples of the brain parenchyma and cerebrospinal fluid of affected patients from PD compared to control samples [8]. Furthermore, several research studies have shown that dopaminergic neurons are particularly sensitive to reactive oxygen species (ROS) due to their reduced antioxidant capacity and increased iron accumulations [9,10]. In recent years, there have been many advances regarding the mechanisms that lead to dysfunction and cell loss in PD. Current therapies aim to reduce inflammation, oxidative stress and mitochondrial dysfunction. However, to date, there is no therapy that delays the neurodegenerative process. 

In recent years, many studies have focused on the analysis of the natural phytocomponents present in foods as important bioactive molecules against chronic age-related diseases, such as neurodegenerative diseases [11,12,13,14,15,16]. In particular, numerous studies support the beneficial effects of the Mediterranean diet (MD) in preventing neurodegeneration. The MD predicts a high intake of cereals, vegetables, fruit, olive oil and legumes, as well as low meat consumption and a moderate amount of fish, seafood and alcohol. This translates into a reduction in the consumption of saturated fat and an increase in the intake of foods containing high concentrations of bioactive compounds, such as polyphenols and flavonoids [17,18]. In this regard, one of the main phytochemicals present in oil and table olives is HT. Its importance is supported by numerous studies that highlight the high antioxidant power of HT and its ability to eliminate free radicals [17]. Therefore, interest in HT has increased in recent years and this is also due to the possession of other biological activities, including anti-inflammatory, antimicrobial, anticarcinogenic and neuroprotective effects [17,19,20,21,22]. All of these activities could give HT a central role in the prevention of neurodegenerative diseases. Although there are still few studies on the neuroprotective effects of HT in animal models, they provide important data that encourage the evaluation of this compound in the treatment of neurodegenerative disorders. In an animal model similar to Huntington’s disease, the antioxidant effect of oral administration of extra virgin olive oil (EVOO) and HT has been shown in the brain. The results show a decrease in lipid peroxidation and an increase in cellular glutathione (GSH) levels. This displays that EVOO and HT act as brain antioxidants through a natural mechanism useful for protection from oxidative damage [23]. In another study, C57BL/6 mice were treated with oligomeric acid Aβ_1–42_ + ibotenic acid to induce neural behavioral dysfunction. After induction, a severe deficiency in their visuo-spatial and working memories took place. However, HT treatment appreciably ameliorated their spatio-cognitive performances. Further research has shown that HT administration has been able to counteract the dysregulation of the signaling mechanisms in hippocampal neurons [24,25,26]. Recently, increasing evidence reports that HT and its derivates activate the phase 2 response, leading to the expression of the nuclear factor erythroid 2-related factor (Nrf2) antioxidant pathway [27,28]. Interestingly, Nrf2 represents a crucial mechanism of resistance to oxidative stress and inflammation in vitro and in vivo [29,30]. In line with these observations, Nrf2 codifies the antioxidant pathway of vitagenes, which exist to counteract various forms of stress (e.g., oxidative, environmental and proteotoxic stress). The latter involves redox-sensitive genes, such as γ-glutamyl cysteine synthetase (γ-GCS), HO-1, heat-shock protein 70 (Hsp70), thioredoxin and sirtuin-1 (Sirt1), called vitagenes, which help preserve the protein homeostasis and cellular redox balance in various pathological states [31,32]. Numerous epidemiological reports suggest a link between pesticides and the incidence of PD [33]. Rotenone, currently used as a fat-soluble insecticide, is able to induce oxidative stress, which leads to the death of dopaminergic (DA) cells [34,35]. Therefore, rotenone seems to reproduce almost all the characteristics of PD, including the accumulation of α-synuclein and the formation of LB [36,37]. Based on this evidence, the aim of our study was to evaluate whether the aqueous extract of the olive pulp containing 40–50% of HT, known as Hidrox^®^ (HD), prevents the neurodegenerative process in an animal model of rotenone-induced PD. In particular, we hypothesize that HD is able to act not only on the oxidative stress through the Nrf2 pathway but also on the neuroinflammation and other important mechanisms involved in the pathogenesis of PD.

## 2. Materials and Methods

### 2.1. Animals

For this study we used CD1 male mice of about 25–30g (Harlan Nossan, Milan, Italy). They were first adapted to their habitat for a week and then were kept in a controlled environment (22 ± 1 °C with a 12 h dark, 12 h light cycle) and had water and food for rodents available. Before the execution of the study, we received the approval of the Review Board of the University of Messina for the care of animals. In addition, the experiments on mice complied with U.S. regulations (Animal Welfare Insurance No. A5594-01, Department of Health and Human Services, Washington, DC, USA), Europe (OJ of EC L 358/1 12/18/1986) and Italy (DM 116192).

### 2.2. Rotenone-Induced PD and Treatment

Eight-week-old male CD1 mice were treated with rotenone or saline. For rotenone intoxication, mice received a daily oral gavage (o.s.) of rotenone (5 mg/kg in 4% carboxymethylcellulose, CMC; Sigma, St. Louis, MO, USA) in saline. For the polyphenol treatment (10 mg/kg), mice received “Hidrox^®^” by intraperitoneal injections (i.p.), daily till the end of the experiment. HD, which was kindly provided by Oliphenol LLC., (Hayward, CA, USA) is a freeze-dried powder prepared from the aqueous portion of olives extracted from the defatted olive pulps, a derivative during the processing of *Olea europaea* L. for olive oil extraction [38]. A total of 12% of the HD extract is made up of polyphenols. Among these, the most abundant in HD is hydroxytyrosol, with 40–50%, while 5–10% is oleuropein, 0.3% is tyrosol and about 20% oleuropein aglycone and gallic acid [39] (Appendix A).

On the 28th day the mice were anesthetized with ketamine (2.6 mg/kg) and xylazine (0.16 mg/kg) and subsequently beheaded. The brains were taken for various analyzes. The doses of rotenone (5 mg/kg) and HD (10 mg/kg) were chosen based on studies in the literature [40,41].

### 2.3. Experimental Group

Animals were casually distributed into the following groups: 

*Group 1*: Sham = vehicle solution (saline) was administrated daily, as rotenone protocol, o.s. (N = 10).

*Group 2*: Sham + HD = HD solution was administrated by i.p. for 28 days (data not shown) (N = 10)

*Group 3*: Rotenone + vehicle = rotenone solution was administrated daily o.s. and vehicle solution (saline) was administrated i.p. daily (N = 10).

*Group 4*: Rotenone + HD = rotenone solution was administrated daily o.s. and HD solution was administrated by i.p. daily for 28 days and 1 h after rotenone administration (N = 10).

### 2.4. Behavioral Testing

Behavioral evaluations on all animals (N = 10 mice for each group) were made 1 day prior and 28 days after the rotenone injection.

### 2.5. Pole Test 

A pole test (PT) was performed to detect motor alteration, such as bradykinesia, as previously described [42,43].

### 2.6. Rotarod Test

Motor activity was assessed with a rotary rod apparatus using a protocol previously described [44,45].

### 2.7. Catalepsy Test

Catalepsy, demarcated as a reduced capability to start movement and a failure to correct posture, was measured as previously described [46,47].

### 2.8. Histology

The brains of 5 mice for each group were fixed, cut and colored with hematoxylin and eosin (H&E, Bio-Optica, Milan, Italy) and subsequently the sections were analyzed using an optical microscope connected to an imaging system (AxioVision, Zeiss, Milan, Italy), as previously described [30,35]. Histopathological evaluation was performed blindly using a semi-quantitative five point rating: 0 = normal, no death neuron observed; 1 = insignificant pathology, SN contained one to five death neurons; 2 = modest pathology, SN contained five to 10 death neurons; 3 = severe pathology, SN contained more than 10 death neurons; 4 = more severe pathology, SN contained only death neurons [48]. The scores from all slides were averaged to give a final score for each individual mouse. Images are representative of all animals in every group.

### 2.9. Stereological Analysis 

Impartial counting of TH-positive DA neurons within the SN was carried out as previously described [49]. The polyclonal rabbit anti-TH antibody (1:400, Merck-Millipore, AB152, Burlington, MA, USA) was used, which was incubated at 4 °C O/N. The sections were treated with the ABC (Vector Labs, Burlingame, CA, USA) method. The images were taken blind, and for each sample, 5 representative sections of the SN were analyzed using StereoInvestigator software (Microbrightfield, Williston, VT, USA).

### 2.10. Immunohistochemical Localization of Tyrosine Hydroxylase (TH), Dopamine Transporter (DAT) and α-Synuclein (α-syn)

The immunohistochemical techniques used have been previously described [50,51]. The antibodies that were incubated O/N on the brain sections were anti-TH (Millipore, 1:500 in PBS, *v*/*v*, AB152, Burlington, MA, USA), anti-DAT (Santa Cruz Biotechnology, 1:300 in PBS, *v*/*v*, 65G10 sc-32258, Dallas, TX, USA) and anti-α-syn (Santa Cruz Biotechnology, 1:50 in PBS, *v*/*v*, LB509 sc-58480, Dallas, TX, USA). To verify the specificity of the antibodies, the brain sections of 5 mice for each group were treated either with a primary or only with a secondary antibody. The images were taken using a Zeiss microscope and Axio Vision software. The ImageJ IHC profiler plug-in was used for densitometric analysis. When this is selected, it automatically traces a histogram profile of the deconstructed DAB image and a corresponding score log is shown [52]. The histogram profile corresponds to the positive pixel intensity value obtained from the computer program [53]. Immunohistochemical analyzes were performed by experienced people who did not know the treatment.

### 2.11. Immunofluorescence Co-localization of TH/α-syn 

Immunofluorescence analysis was performed with the protocol that we previously described [50,51]. The antibodies that were incubated in a humidified chamber at 37 °C O/N on the brain sections of 5 mice for each group were anti-TH (1:250; Merck-Millipore, AB152, Burlington, MA, USA) and anti-α-syn (1:50; Santa Cruz Biotechnology, LB509 sc-58480, Dallas, TX, USA). The sections were analyzed using a fluorescence microscope (Leica DM2000, Wetzlar, Germany). Each photograph taken was digitized with an 8-bit resolution in an array of 2560 × 1920 pixels. The images obtained were subsequently cut out and prepared for the assembly of the figures using Adobe Photoshop 7.0 (Adobe Systems, Palo Alto, CA, USA).

### 2.12. Western Blot Analysis for IkB-α, NF-kB, Bax, Bcl-2, iNOS, NRLP3, ASC, Caspase-1, IL-18, IL-1β, Hsp70, Sirt-1 and HO1 

Western blot analysis was performed on brains of 5 mice for each group with the protocol that we previously described [54,55]. The levels of IkB-α, Bax, Bcl-2, iNOS, NRLP3, ASC, Caspase-1, IL-18 and IL-1β were quantified in cytosolic, while the NF-kB p65 levels were quantified in nuclear fraction. The specific primary antibodies that were incubated at 4 °C O/N were anti-IkB-α (1:500; Santa Cruz Biotechnology, C-21: sc-371, Dallas, TX, USA), anti-NF-kB p65 (1:500; Santa Cruz Biotechnology, F-6: sc-8008), anti-Bax (1:500; Santa Cruz Biotechnology, P-19: sc-526), anti-Bcl-2 (1:500; Santa Cruz Biotechnology, N-19: sc-492, Dallas, TX, USA), anti-iNOS (1:1000; Transduction Laboratories, 610432), anti-NRLP3 (1:500; Santa Cruz Biotechnology, sc-66846, Dallas, TX, USA), anti-ASC (1:500; Santa Cruz Biotechnology, N-15: sc-22514-R, Dallas, TX, USA) anti-Caspase-1 p20 (1:1000; Santa Cruz Biotechnology, G-19: sc-1597, Dallas, TX, USA), anti-IL-18 (1:500; Santa Cruz Biotechnology, H-173: sc-7954, Dallas, TX, USA), anti-IL-1β (1:500; Santa Cruz Biotechnology, H-153: sc-7884, Dallas, TX, USA), anti-Hsp70 (1:500; Santa Cruz Biotechnology, 3A3 sc-32239, Dallas, TX, USA), anti-Sirt1 (1:500; Santa Cruz Biotechnology, B7 sc-74465, Dallas, TX, USA), anti-HO-1 (1:500; Santa Cruz Biotechnology, A3 sc-136960, Dallas, TX, USA) and anti-γGCs (1:500; Santa Cruz Biotechnology, H5 sc-390811, Dallas, TX, USA). Protein lysates were also incubated with a β-actin antibody or laminin antibody (1:5000; Santa Cruz Biotechnology, C4 sc-47778, E1 sc-376248, Dallas, TX, USA), in order to verify that all samples had been loaded in equal quantities. The signals were captured with BIORAD ChemiDocTM XRS + software thanks to the use of a reagent that emits chemiluminescence (Super Signal West Pico, Pierce chemiluminescent substrate). The relative expression of the bands was subsequently normalized to the β-actin levels. Image analysis was performed using Image Quant TL software, v2003.

### 2.13. Protein Carbonyl Assay

Oxidized proteins were analyzed on tissues of 5 mice for each group by means of the OxyBlotTM Protein Oxidation (Merck Millipore, Darmstadt, Germany), conceding to the manufacturer’s instructions. Briefly, 15 μg protein of each sample were denatured by adding 5 μL of 12% SDS (Sigma-Aldrich, St. Louis, MO, USA). Samples were derivatized by adding 10 μL of a 1 × 2,4 dinitrophenolhydrazine (DNPH, Sigma-Aldrich, St. Louis, MO, USA) solution and incubated for 15 min at RT. Subsequently, samples were neutralized with 7.5 μL of a neutralization solution and the proteins were separated by SDS/PAGE. The primary antibody used was against DNPH and revealed by luminescence (SuperSignal detection system kit: Pierce Chemical, Dallas, TX, USA). The bands were quantified, normalizing the pixels in each lane to the loading control band (Gel-Logic 2200-PRO Bioscience, London, UK), and analyzed (Molecular Imaging software).

### 2.14. Statistical Evaluation 

In the figures and in the text the values are expressed as the average ± SEM and are representative of at least 3 experiments carried out at different times. Ten mice per group were used in each experiment unless otherwise noted. The data review was performed by one-way analysis of variance followed by a Bonferroni post-hoc test for multiple comparisons. A *p* value of less than 0.05 was considered significant. * *p* < 0.05; ** *p* < 0.01; *** *p* < 0.001. 

## 3. Results

### 3.1. Effect of HD Treatment on Behavioral Impairments and on the Neuronal Degeneration of the Dopaminergic Tract Induced by Rotenone Administration

To investigate the relationship between the degeneration of dopaminergic neurons, rotenone-induced, and the recovery processes, we analyzed the motor activity 1 day prior and 28 days after the rotenone induction. The data at time Point 0 are not shown as no significant differences between the different groups were observed. The pole test was used to assess whether the rotenone-induced mouse model efficaciously induced bradykinesia [56]. “Time to turn” and “Total time” notably increased following injection of rotenone compared with the Sham group (Figure 1A,A1). HD treatment significantly reduced “Total time” (45%) and “Time to turn” (61%) (Figure 1A,A1), suggesting a substantial reduction of bradykinesia. In addition, through the Rotarod test we evaluated the motor function. A total of 28 days after the induction of PD, the mice showed significant motor changes evidenced by the reduction in the time spent on the Rotarod and by the greater number of falls. In contrast, HD-treated mice had a significant reduction in motor deficits (Figure 1B). Moreover, the rotenone produced an important cataleptic effect in mice. In fact, at 28 days after rotenone injection, the mice exhibited a significant increase in cataleptic symptoms. Otherwise, the daily HD administration significantly reduced the catalepsy duration induced by rotenone (Figure 1C). 

To evaluate the histopathological alteration induced by the administration of rotenone, H&E staining was performed on the brain sections. The mice treated with saline or only with HD for 28 days showed a normal brain architecture and a normal number of neurons in the SN (Figure 1D,E,H). Instead, mice treated with rotenone showed evident alterations, such as cytoplasmic vacuolization, vascular degeneration and nigrostriatal neuronal cell loss. The architecture of the brain in the rotenone mice was altered compared with the control mice (Figure 1F,H). In contrast, mice treated with rotenone and HD showed a marked reduction in vascular degeneration and cytoplasmic vacuolization. It also reduced the cell with pyknotic nuclei in the SN (Figure 1G,H).

### 3.2. HD Treatment Reduced the Loss of TH, DAT and α-Synuclein Expression in the SN Induced by Rotenone Administration

The expression of TH and DAT was evaluated to prove the effect of the HD treatment on the DA pathway. The group of animals treated only with rotenone showed a significant loss of TH-positive cells in the SN (Figure 2B,D) compared to the control group (Figure 2A,D). Instead, HD administration has been shown to significantly reduce the loss of TH-positive neurons in SN (Figure 2C,D). Stereology analysis of nigral TH-positive neurons displayed important neuroprotection by HD treatment. We observed a considerably decline in the number of TH-positive neurons in mice after rotenone injection (Appendix A), compared to Sham group (Appendix A). This loss decreased following HD treatment (Appendix A). Similarly, Nissl-stained neurons were depleted significantly by rotenone lesioning but not in mice that had been treated with HD (Appendix A).

Furthermore, we showed a critical loss of DAT in the rotenone-injected mice at the level of the midbrain (Figure 2F,H) compared to the Sham group (Figure 2E,H), while the HD treatment considerably restored the levels of DAT (Figure 2G,H).

α-Syn is the highest constituent of the intraneuronal protein aggregates known as Lewy bodies. Since the accumulation of α-syn is a characteristic of PD, we wanted to evaluate the expression of this protein in order to determine the ability of HD to counteract the neurodegenerative process. We observed an important immuno-reactivity in the rotenone-injured mice (Figure 2J,L) compared to Sham group (Figure 2I,L). Instead, the treatment with HD notably reduced the α-synuclein expression in the SN after rotenone intoxication (Figure 2K,L).

Furthermore, to prove that the accumulation of α-synuclein occurred in the dopaminergic neurons, we carried out, by immunofluorescence analysis, a double coloring between TH (green) and α-syn (red). In the Sham group we did not find α-syn in TH-positive dopaminergic neurons (Figure 3G), while after rotenone intoxication there was an increase in the accumulation of α-syn in TH-positive neurons (Figure 3H). HD treatment prevented α-syn aggregation in the dopaminergic neurons (Figure 3I).

### 3.3. Effect of HD on Cellular Stress Response after Rotenone Administration 

Oxidative stress in the brain is accompanied by augmented expression of genes contributing to the free radicals’ detoxification, rescue of mitochondrial function and cell survival stress responsive genes called vitagenes. Vitagenes encode for the Hsps, HO-1 and sirtuin protein systems, whose activation is most likely dependent on Nrf2. In agreement with previously reported data from our laboratory, demonstrating the beneficial effects of HD against PD [27], we wanted to evaluate the effect of HD treatment on the activation of Nrf2 and consecutively on HO-1, Hsp-70 and Sirt-1 expression. Western blot analysis revealed a significant increase in Nrf2 in rotenone-treated mice. Treatment with HD further increased the levels of this protein (Figure 4A). In addition, we demonstrated that HO-1, Hsp-70 and Sirt-1 expression also began to increase in mice treated with rotenone, but we observed more significant levels of this protein in the HD-treated mice (Figure 4C–E). In addition, we investigated rotenone’s effect in the absence or presence of HD in different areas of the brain. Levels of Hsp70, measured by Western blot analysis, in the cortex (CX), substantia nigra (SN), striatum (ST) and hippocampus (HP) in the rotenone-treated animals—in the absence and presence of HD administration and compared to the untreated control animals—are shown in Figure 5. Treatment of the rotenone-injected animals with HD resulted in a significant increase in the synthesis of Hsp70 in all the brain regions examined, whereas this effect was not observed in the group of mice receiving rotenone alone, as compared to the controls, in all brain regions examined, except for SN, where the rotenone-induced increase in Hsp70 was significant vs. both the control and rotenone plus HD treatment groups. A representative Western blot, obtained probing the different brain regions for Hsp70 proteins, is shown in the same figure (Figure 5A–D1). We also examined the regional expression of γ-GGCS in the same experimental conditions. Rotenone administration, given alone, significantly increased the γ-GCS expression in the brain regions of SN and striatum, but not in the cortex or hippocampus, as compared to the control group of animals. In response to the HD administration, the animals with the damage showed increased synthesis of the γ-GCS protein, as compared to controls, and this increase was higher than the levels found in the group of mice receiving only rotenone, in all brain regions examined. A representative Western blot, obtained probing the brain homogenate, obtained from a specific region with an antibody specific for the γ-GCS protein, is also shown (Figure 6A–D1).

### 3.4. Modulation of Protein Carbonyls in Mice Brain after Rotenone Treatment and HD Supplementation

Following oxidative stress, the accumulation of oxidation products of the proteins and lipids occurs, which are measured, respectively, by protein carbonyl and 4-hydroxynonenal (HNE) [57,58]. The formation of carbonyl groups in the residues of amino acids and HNE from arachidonic acid or other unsaturated fatty acids, following the oxidation of proteins and lipids, represents a clear sign of oxidative attack of free radicals and damage to proteins and lipids [57]. Protein carbonylation and HNE exerts negative effects on cell function and viability, being generally unrepairable and leading to production of potentially dangerous protein aggregates and to cellular dysfunction [57]. Examination of the brain protein carbonyls in the group of rotenone-treated mice revealed a significant elevation in all brain regions examined with respect to the control group, while in the rotenone-treated animals supplemented with HD, we observed a significant reduction of protein carbonyl (Figure 7A–C). 

### 3.5. Effect of HD Treatment on NF-κB, IκB-α, iNOS Expression and on Apoptosis Induced by Rotenone Administration

In order to evaluate the anti-neuroinflammatory effect by which HD treatment may attenuate the development of PD, we investigated the expression of NF-κB and IκB-α by Western blot analysis in the midbrain samples. NF-κB nuclear translocation was significantly increased in the rotenone group, while the HD treatment attenuated NF-κB expression (Figure 8A). IκB-α degradation was lowered in rotenone-injured mice vis-a-vis the Sham mice, while the HD treatment increased IκB-α cytosolic activity (Figure 8C). In addition, to determine the role of nitric oxide (NO), iNOS expression was assessed 28 days after rotenone administration. The analysis evidenced a significant increase in the iNOS levels in the SN of the rotenone group, which was considerably reduced by HD treatment (Figure 8D). In order to evaluate the effect of the HD treatment on the rotenone-induced apoptosis we analyzed the expression of Bax and Bcl-2. Tissues collected from the rotenone-treated animals showed an increased Bax expression, compared to the control mice. HD treatment reduced this expression (Figure 8E). Samples taken from Sham mice showed basal levels of Bcl-2; rotenone administration reduced this expression. HD treatment restored Bcl-2 expression to the basal levels (Figure 8F).

### 3.6. Effect of HD Treatment on Inflammasome Pathway and Caspase-1, IL-1β, IL18 Expression Induced by Rotenone Administration

To evaluate the effect of the HD treatment on the activation of the inflammasome pathway, we investigated by Western blot analysis the levels of expression of NRLP3 and ASC. We showed an increased NRLP3 expression in the rotenone-treated group compared to the Sham mice, and a downregulation of this expression in samples collected from the HD-treated mice (Figure 9A). Western blot analysis revealed an upregulation of the ASC levels in samples collected from rotenone-treated mice, compared to the control group. HD-treated mice showed an inhibition of the ASC expression (Figure 9B). Brain tissues from rotenone-treated mice showed an increased expression of Caspase-1 compared to the control mice. HD treatment reduced this expression (Figure 9C). Western Blot analysis displayed an upregulation of IL-1β levels compared to the Sham animals. HD administration downregulated IL-1β expression (Figure 9D). HD treatment also reduced IL-18 expression induced by rotenone-administration. Samples collected from Sham mice showed the basal expression of IL-18 (Figure 9E).

## 4. Discussion

The Mediterranean diet (MD) has numerous advantageous effects in preserving health, especially during aging or neurodegeneration [59]. Notably, many recent data highlight the importance of the phenols contained in EVOO to counteract the misfolding and toxicity of proteins, with particular attention to the mechanisms that lead to the initiation and progression of PD [60,61]. The main component of the EVOO and olive is hydroxytyrosol, which represents an important protective factor. HT has proven to have the highest antioxidant activity ever measured compared to any other known natural antioxidant [62,63]. HT, in addition to reducing oxidative stress by inhibiting free radicals and eliminating reactive oxygen and nitrogen, has been shown to have anti-inflammatory and anti-microbial effects in humans and animals, and stimulate the immune system [64,65,66]. Hidrox^®^ is an aqueous extract of olive containing 40–50% HT. In this compound, thanks to a unique manufacturing process, all the essential elements, such as HT, remain intact in their natural matrix. In addition, several in vitro and in vivo studies confirmed that HD shows higher antioxidant and anti-inflammatory activity, as demonstrated by electron spin resonance (ESR) spectroscopic analysis after oxidation of lipids of the mitochondrial membrane by free radicals [67,68]. HD properties have also been evaluated in an in vitro study of neuroinflammation, which highlighted the ability of this compound to reduce the production of pro-inflammatory cytokines, such as TNF-α, IL-1β and IL-6 [69]. Fascinatingly, our recent in vivo studies with olive polyphenols have demonstrated that oleuropein and its derivates, especially HD, exerts neuroprotective effects, an improved overall of health span and longevity as well as stress resistance in a rotenone-stressed PD model of *C. elegans* [70].

Here, thanks to our PD model induced by rotenone on male mice, we have shown that administration of HD during the entire period of the PD induction by rotenone rises the redox potential, correlating with induction of vitagenes, and helps susceptible neurons to resist to proteotoxic insults and therefore neurodegeneration. Restoration of normal proteostasis seems to be crucial for neuronal survival. Therefore, our study does not aim to evaluate a treatment for the cure of PD but to demonstrate that the use of natural compounds, such as HD, could prevent the neurodegenerative process typical of this pathology. Several statistical studies have shown that more men than women are diagnosed with PD [71]. The differences between gender of symptoms, course and cognitive aspect have not yet been extensively examined. A source of cognitive differences could be the effect of estrogen on the dopaminergic neurons and pathways in the brain. For this reason, we decided to carry out our research on male mice, but the study on females would also be interesting. Our results have shown that HD is able to act not only on oxidative stress but also on the inflammatory response, apoptosis and inflammasomes, thus containing the accumulation of α-synuclein, the loss of dopaminergic neurons and the behavioral deficits (Appendix A).

Nrf2, one of the most important transcription factors that activates several genes with cytoprotective function, participates in antioxidant and anti-inflammatory reactions [72,73,74,75,76,77]. These cytoprotective genes encode a wide variety of phase II detoxification enzymes, such as NAD (P) H: quinone oxidoreductase 1 (NQO1), HO-1, γ-GCS, glutathione S-transferase (GST) and glutamate-cysteine ligase, thioredoxin, thioredoxin reductase, thermal shock proteins and many others [31,78,79,80,81]. Cell culture studies have shown that HT was the only olive oil phenol capable of increasing the transactivation of Nrf2, which suggests that this polyphenol could be responsible for the induction of Nrf2-dependent gene expression [82]. Therefore, our hypothesis is that HD carries out its neuroprotective action through the activation of the Nrf2/HO-1 axis.

As an effect of oxidative stress in tissues and organs, protein and lipid oxidation occur. Introduction of carbonyl groups into amino acid residues is a hallmark for oxidative injury to proteins by ROS. Protein carbonylation can have harmful effects on cell function and viability, since it is generally unrepairable by cells and can lead to protein dysfunction and to the production of potentially harmful protein aggregates [57]. Analysis of the carbonyls in the brain proteins revealed significantly higher levels in the rotenone group compared to that treated with HD. Moreover, our results showed the ability of HD to significantly increase the levels of Nrf2, HO-1, Hsp-70, Sirt-1 and γ-GCS compared to the group treated only with rotenone.

As observed in several studies [83,84], and confirmed by our results, we can indicate a relationship between Nrf2 and NF-kB, a transcription factor responsible for the inflammatory response that activates many genes coding for pro-inflammatory cytokines and immunoregulatory mediators.

Here we have shown that HD is capable of preventing the nuclear translocation of NF-κB and the degradation of IkB-α. In addition, the iNOS levels have also been significantly reduced by HD. This suggests that regulation of redox homeostasis by Nrf2 probably leads to modulation of NF-kB activity and the inflammatory response characteristic of PD. Another complex that could be modulated by Nrf2 is the NLRP3 inflammasome, whose activation is linked to various stressors. Therefore, in our study we evaluated the Caspase-1, IL-1β, IL-18, ASC and NLRP3 levels by demonstrating that HD was actually able to significantly reduce the expression of these proteins, which was instead increased by rotenone.

Our study also showed that HD protects neuronal cells from rotenone-induced apoptosis. The levels of the Bax pro-apoptosis marker were significantly reduced by HD; conversely, the anti-apoptotic protein Bcl-2 was significantly increased.

Overall, the modulation of the processes just described allowed a reduction in the death of dopaminergic neurons evidenced by the restoration of the levels of two specific markers, TH and DAT, in the brains of HD-treated mice. In addition, HD prevented the α-synuclein from aggregating and forming accumulations in the dopaminergic neurons, which is also reflected in the reduction in the motor deficits induced by rotenone.

## 5. Conclusions

In conclusion, our results have shown that Hidrox^®^ is a very effective antioxidant and a powerful anti-inflammatory agent. Therefore Hidrox^®^ represents an effective nutritional product that could be used as a preventive agent in the neurodegenerative process characteristic of PD. However, further studies are needed to clarify whether Nrf2 is activated following Keap1 dissociation due to oxidative stress, the modification of the thiol groups of Keap1, or due to the post-translational modifications mediated by phosphatidylinositol 3-kinase (PI3K), c-Jun N-terminal kinase (JNK), protein kinase C (PKC) and protein kinases regulated by extracellular signals (ERK). It would also be interesting to evaluate whether HD maintains its neuroprotective properties regardless of sex.

## Figures and Tables

**Figure 1 antioxidants-09-00824-f001:**
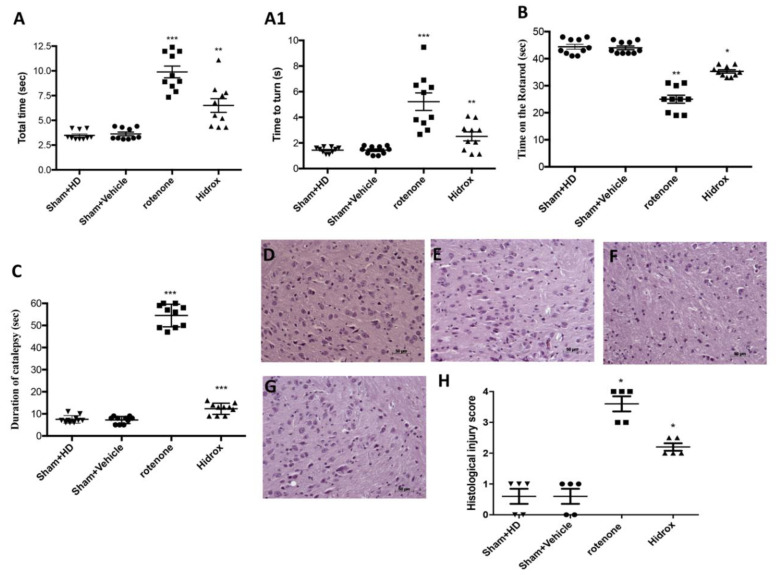
Effect of HD on behavioral impairments and on histological parameters induced by rotenone intoxication. (**A**,**A1**) Motor function was assessed using a Pole test. At 28 days, mice exhibited a significant motor dysfunction as indicated by an increase in “Time to turn” and “Total time” spent to descend to the floor following injection of rotenone compared with the Sham group. HD administration notably reduced “Total time” and “Time to turn”. (**A**) *** *p* < 0.001 vs. Sham; ** *p* < 0.01 vs. rotenone; (**A1**) *** *p* < 0.001 vs. Sham; ** *p* < 0.01 vs. rotenone. (**B**) At 28 days, using a Rotarod apparatus, mice exhibited a significant motor dysfunction as indicated by a decrease in time spent on the Rotarod. HD treatment blunted the motor dysfunction in mice. (**B**) ** *p* < 0.01 vs. Sham; * *p* < 0.05 vs. rotenone. (**C**) Catalepsy was evaluated according to the standard bar hanging procedure; this motor test showed that the HD treatment reduced behavioral impairment induced by rotenone. (**C**) *** *p* < 0.001 vs. Sham; *** *p* < 0.001 vs. rotenone. Values are the mean ± SEM of 10 mice for each group. Sham+HD and Sham+vehicle groups showed no evidence of degenerating cells in the SN (**D**,**E**), whereas degeneration of neuromelanin-pigmented cells was evident in the SN of the rotenone-treated animals (**F**). HD treatment restored the architecture compared to the control mice (**G**). The data are representative of at least three independent experiments and are expressed as the mean ± SEM of 5 mice for each group. (**H**) * *p* < 0.05 vs. Sham; * *p* < 0.05 vs. rotenone. Scale bar: 50 μm.

**Figure 2 antioxidants-09-00824-f002:**
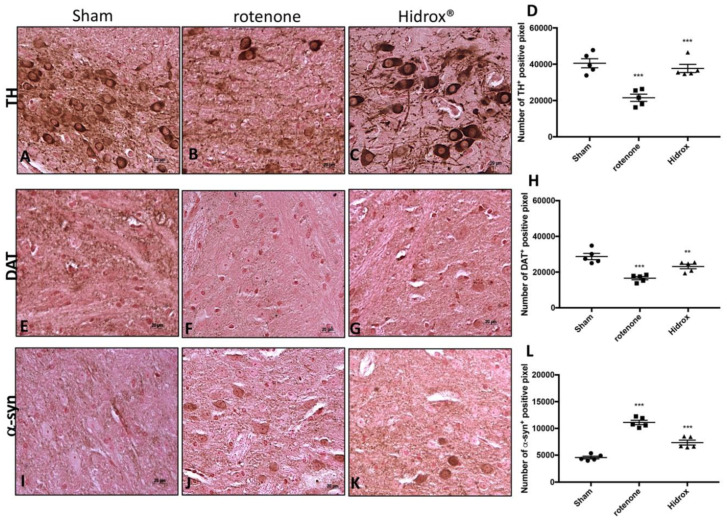
Effects of HD on TH, DAT and α-synuclein expression in SN of rotenone-treated mice. The immunohistochemical analysis has shown, compared with the Sham mice (**A**), a noticeable loss of TH-positive cells (**B**). Animals treated with HD revealed an increase in TH expression (**C**). *** *p* < 0.001 vs. Sham; *** *p* < 0.001 vs. rotenone (**D**). The immunohistochemical analysis revealed, compared with Sham group (**E**), an evident loss of DAT-positive cells (**F**). Animals subjected to treatment with HD revealed a positive stain for DAT (**G**) compared with rotenone group. *** *p* < 0.001 vs. Sham; ** *p* < 0.01 vs. rotenone (**H**). The midbrain was marked with antibodies against α–synuclein aggregation (**I**–**K**). The immunohistochemical analysis revealed, compared with Sham animals (**I**), a positive staining for α-synuclein (**J**). HD treatment appreciably reduced the positive staining for α-synuclein in the SN (**K**). *** *p* < 0.001 vs. Sham; *** *p* < 0.001 vs. rotenone (**L**). Data are expressed as the % of TH-positive pixels and are the means ± SEM of 5 mice/group. Scale bar: 20 μm.

**Figure 3 antioxidants-09-00824-f003:**
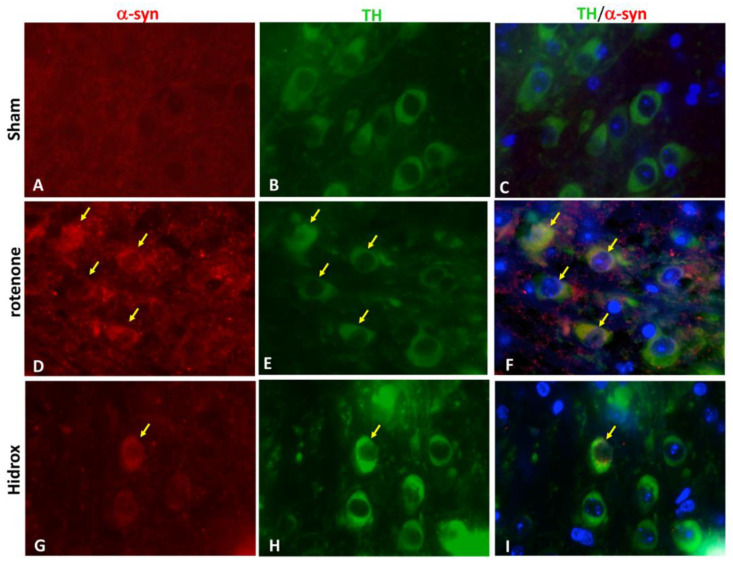
Effects of HD on α-synuclein expression and a double staining of TH/α–synuclein after rotenone-intoxication. Results are shown for Sham (**A**–**C**), mice after rotenone-intoxication (**D**–**F**) and mice treated with HD (**G**–**I**). Midbrain sections were double stained with antibodies against TH (**B**,**E**,**H**—green)/α-syn (**A**,**D**,**G**—red). Midbrain sections revealed an important expression of α-syn in dopaminergic neurons in the rotenone group (**F**), compared with Sham group (**C**). TH/α-syn, double stained, was decreased after HD treatment (**I**). The photographs are illustrative of at least three experiments performed on different days, and are representative of all animals in every group (N = 5 mice/group). Images were digitalized at a resolution of 8 bits into an array of 2048 × 2048 pixels. Scale bar: 100 μm.

**Figure 4 antioxidants-09-00824-f004:**
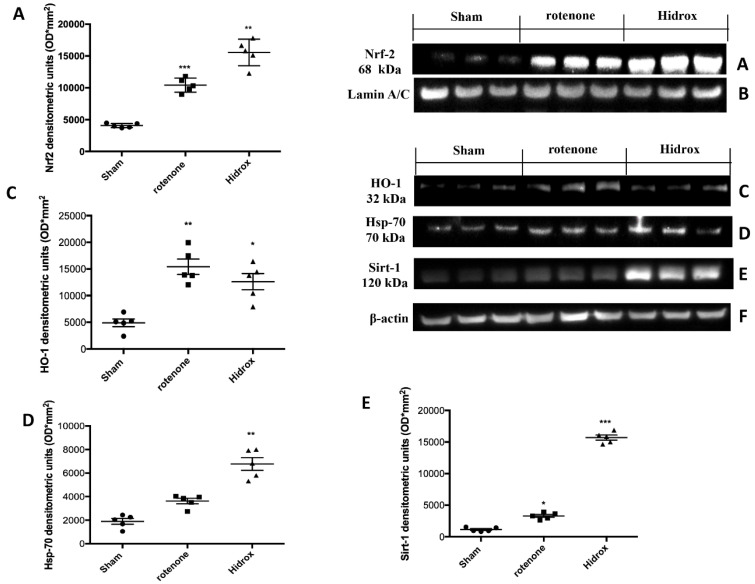
Effects of HD on Nrf2, HO-1, Hsp-70 and Sirt-1 proteins after rotenone intoxication. Western blot analysis demonstrated Nrf2 expression and Sirt-1 expression to be significantly increased in the rotenone group, whereas treatment with HD significantly further increased Nrf2 expression (**A**). HO-1 expression was increased after rotenone intoxication; treatment with HD decreased the levels of this protein (**C**). Hsp-70 expression was increased after rotenone intoxication; treatment with HD maintained high levels of this protein (**D**). Sirt-1 expression was increased after rotenone intoxication; treatment with HD decreased levels of this protein (**E**). Protein lysates were also incubated with a β-actin antibody (**F**) or laminin antibody (**B**) in order to verify that all samples had been loaded in equal quantities. The data are expressed as the mean ± SEM from N = 5 mice/group. (**A**) *** *p* < 0.001 vs. Sham; ** *p* < 0.01 vs. rotenone; (**C**) ** *p* < 0.01 vs. Sham; * *p* < 0.05 vs. rotenone; (**D**) ** *p* < 0.01 vs. rotenone; (**E**) * *p* < 0.05 vs. Sham; *** *p* < 0.001 vs. rotenone.

**Figure 5 antioxidants-09-00824-f005:**
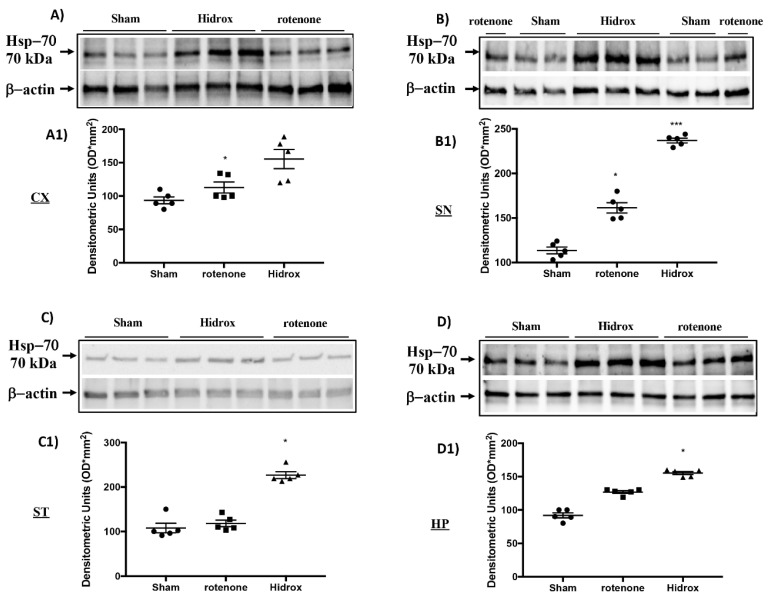
Effects of HD on the Hsp-70 protein after rotenone intoxication. Western blot analysis showed that the expression of Hsp-70 increased insignificantly in the rotenone group in CX (**A**,**A1**), ST (**C**,**C1**) and HP (**D**,**D1**) regions, while there was a significant increase in the SN (**B**,**B1**) area. HD treatment further significantly increased the expression of Hsp-70 (**A**–**D1**). The data are expressed as the mean ± SEM from N = 5 mice/group. (**A1**) * *p* < 0.05 vs. rotenone; (**B1**) * *p* < 0.05 vs. rotenone; *** *p* < 0.001 vs. Sham; (**C1**) * *p* < 0.05 vs. rotenone; (**D1**) * *p* < 0.05 vs. rotenone.

**Figure 6 antioxidants-09-00824-f006:**
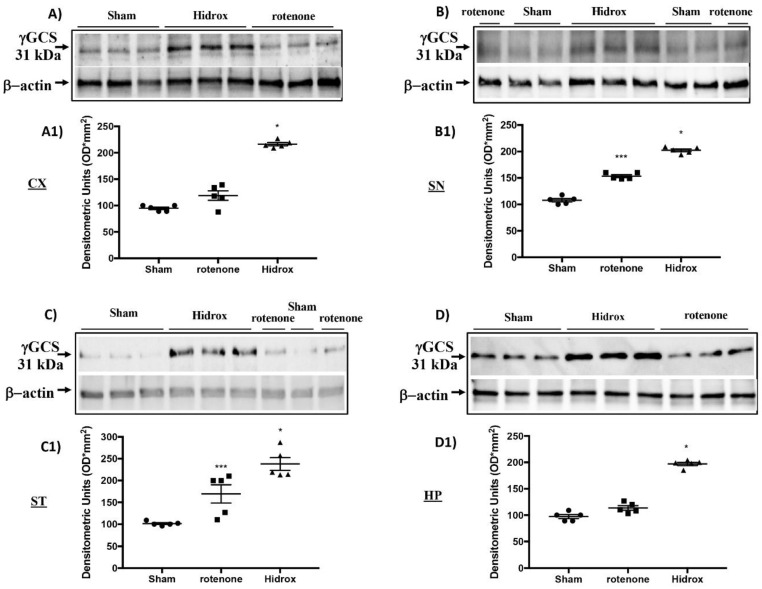
Effects of HD on the γ-GCS protein after rotenone intoxication. Western blot analysis showed that the expression of γ-GCS increased insignificantly in the rotenone group in the CX (**A**,**A1**) and HP (**D**,**D1**) areas, while there was a significant increase in the ST (**C**,**C1**) and SN (**B**,**B1**) regions. HD treatment further significantly increased the expression of Hsp-70 (**A**–**D1**). The data are expressed as the mean ± SEM from N = 5 mice/group. (**A1**) * *p* < 0.05 vs. rotenone; (**B1**) * *p* < 0.05 vs. rotenone; *** *p* < 0.001 vs. Sham; (**C1**) * *p* < 0.05 vs. rotenone; *** *p* < 0.001 vs. Sham; (**D1**) * *p* < 0.05 vs. rotenone.

**Figure 7 antioxidants-09-00824-f007:**
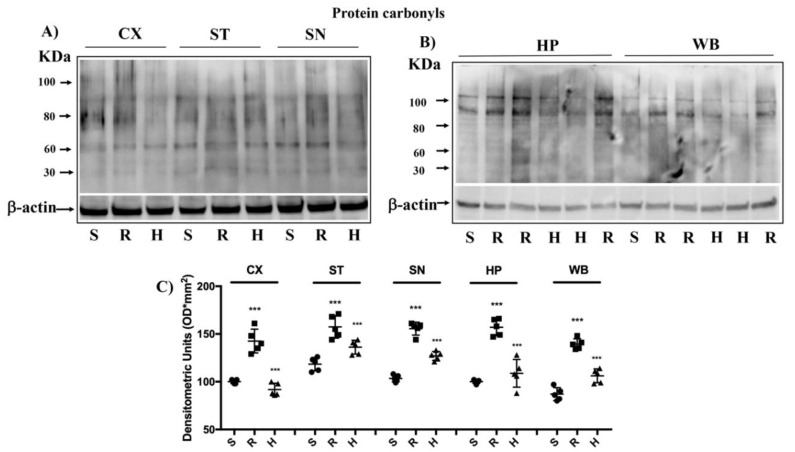
Brain protein carbonyls analysis after rotenone intoxication. (**A**,**B**) The examination of the carbonyls of brain proteins showed a significant increase of these in all brain regions examined after induction of rotenone (R), compared to the control (S) and animals treated with HD (H). The data are expressed as mean ± SEM from N = 5 mice/group. (**C**) *** *p* < 0.001 vs. Sham; *** *p* < 0.001 vs. *rotenone*.

**Figure 8 antioxidants-09-00824-f008:**
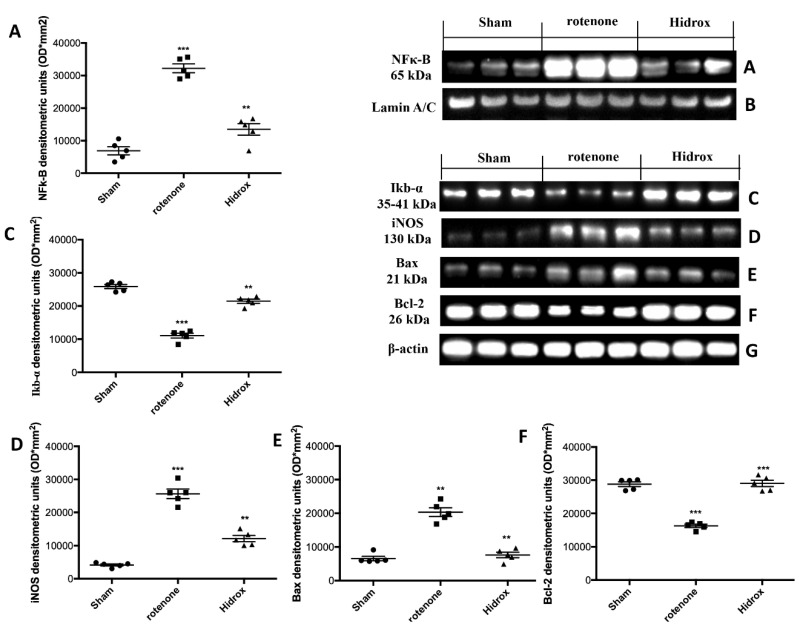
Effects of HD on expression of IκB-α and nuclear translocation of NF-κB p65 and on iNOS after rotenone intoxication. NF-kB levels were significantly increased in the nuclear fraction compared to the Sham animals (**A**). HD treatment significantly reduced NF-kB translocation (**A**). Basal levels of IkB-α were found tissues from Sham group; they were notably reduced in samples from the rotenone-treated mice (**C**). HD administration significantly reduced IkB-α degradation (**C**). Western blot analysis of tissue lysates from rotenone-treated mice shows significant increases in iNOS expression after 28 days. HD significantly lowered the expression of iNOS in the SN after rotenone intoxication (**D**). Western blot analysis demonstrated Bad expression to be significantly increased in the rotenone group, whereas treatment with HD significantly limited the rise in Bad expression (**E**). Finally, Bcl-2 expression was reduced after rotenone intoxication; however, treatment with HD restored the basal levels (**F**). Protein lysates were also incubated with a β-actin antibody (**G**) or laminin antibody (**B**) in order to verify that all samples had been loaded in equal quantities. The data are expressed as the mean ± SEM from N = 5 mice/group. (**A**) *** *p* < 0.001 vs. Sham; ** *p* < 0.01 vs. rotenone; (**C**) *** *p* < 0.001 vs. Sham; ** *p* < 0.01 vs. rotenone; (**D**) *** *p* < 0.001 vs. Sham; ** *p* < 0.01 vs. rotenone; (**E**) ** *p* < 0.01 vs. Sham; ** *p* < 0.01 vs. rotenone; (**F**) *** *p* < 0.001 vs. Sham; *** *p* < 0.01 vs. rotenone.

**Figure 9 antioxidants-09-00824-f009:**
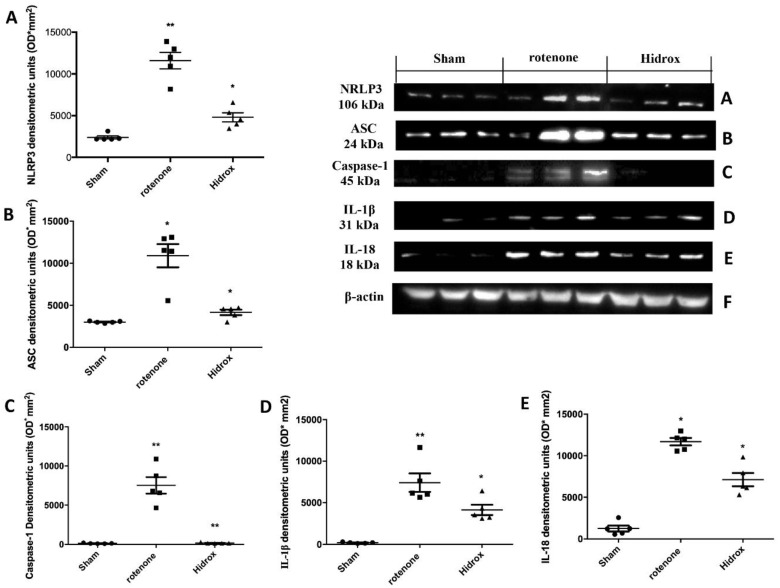
Effects of HD on the inflammasome pathway after rotenone administration. Brain tissues collected from the rotenone-treated animals displayed increased levels of NRLP3 expression compared to the Sham mice (**A**). HD treatment reduced NRLP3 expression (**A**). Western blot analysis also showed an increased expression of the ASC levels in rotenone-treated mice compared to the control mice (**B**). HD treatment decreased ASC expression (**B**). Western blot analysis showed significant expression levels of Caspase-1 in SN of rotenone-treated mice, which were reduced significantly by HD (**C**). The increase in IL-1β expression observed in the rotenone group was likewise limited to a significant extent with HD (**D**). Furthermore, Western blot analysis displayed an upregulation of IL-18 expression compared to the Sham mice. HD administrations notably reduced this expression (**E**). Protein lysates were also incubated with a β-actin antibody (**F**) in order to verify that all samples had been loaded in equal quantities. The data are expressed as the mean ± SEM from N = 5 mice/group. (**A**) ** *p* < 0.01 vs. Sham; * *p* < 0.05 vs. rotenone; (**B**) * *p* < 0.05 vs. Sham; * *p* < 0.05 vs. rotenone; (**C**) ** *p* < 0.01 vs. Sham; ** *p* < 0.01 vs. rotenone; (**D**) ** *p* < 0.01 vs. Sham; * *p* < 0.05 vs. rotenone; (**E**) * *p* < 0.05 vs. Sham; * *p* < 0.05 vs. rotenone.

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
