# Peer review of "Anti-inflammatory and Anti-oxidant Activity of Hidrox® in Rotenone-Induced Parkinson’s Disease in Mice"

_antioxidants, 2020, doi:10.3390/antiox9090824_

Round 1

Reviewer 1 Report

In my opinion, authors must rewrite the paper avoiding reuse significant portions of their previously published work.

Author Response

Thank you for your time and your review report

Reviewer 2 Report

The article deals with an interesting subject which is the potential neuroprotective effect of Hidrox® in an animal model of Parkinson’s disease. The study provides remarkable evidences on the ability of this extract to activate the adaptive stress pathway in the potential treatment of degenerative diseases. 

In my opinion, the study is well designed in order to achieve the objectives and the methodology is correct. However, Figures should be arranged to appropriately and clearly illustrate the findings.

In Material and Methods section, the description of distribution of samples from animals for histological and Western blot techniques must be clarified. It is not clear if a total of 10 animals for each technique are used or samples from the same 10 animals are distributed for both of them. In histology, semi-quantitative rating is defined, indicating ‘Representative images are shown’. But those images are not included in the Manuscript. They should be. Moreover, in experimental group, Group 2 (Sham + vehicle) is described, but data are not shown. This should be justified. Or maybe, the group 2 should be excluded from this study…

Concerning Results section, the Authors have used different symbols (* and #) in order to distinguish comparisons between groups, but as they consist of statistical symbols, the universal nomenclature should be appropriately used in all of them.

In Lines 314 and 315, ‘cellular inflammation’ and ‘invisible nuclei’ are not correct terms. They must be substituted by the pathological technical terminology.  

All immunohistochemical images (not confocal) should be edited by experts in order to illustrate the findings with the real colours. Furthermore, groups from which each sample comes and marker used must be clearly indicated in the panel. It is really hard to understand the results provided by means of the Figures at present configuration.   

In Figures 4 and 13 only two groups are compared while three are in the rest of Figures.

In Figure 5, Lines 391 and 392 are referring to E – G instead of A – C. 

Discussion section is appropriate. Only in Conclusions, the last sentence should be re-written in order to reflect the real impact of the findings. It sounds too much ambitious and it could even raise excessive and unreal expectations. 

Some minor points:

  • In Material and Methods section, commercial reference details are missing for some reagents used.
  • In Figure S1, the abbreviation GC-MS must be defined.
  • Some spaces and words appear duplicated (for instance, line 128 …..of of HD….)

Author Response

  1. Methods (Line 136): “Group 2: Sham + HD = HD solution was administrated by i.p. for 28 days (data not shown; N=10)”.  Why is this group not shown? For some of the presented data, such as Figs 10-12, it is critical to investigate whether HD alone has the same effect as HD+Rotenon.

As suggested by the referee, we included in the histological results the Sham+HD group to demonstrate that there are no differences with the Sham group.

  1. Individual data points need to be shown on bar graphs on all figures. Was the number of samples always 10/group on all the figures? If not, indicate N in figure legends.

Thank you for your suggestion, we inserted in the figure legends the number of the animals that we used for the analysis.

Other:

  1. Methods (Line 140): Please, provide more details about drugs administration. Was HD injected before rotenone? Was there a delay between the second drug delivery?

Thank you for your suggestion. We have included in the experimental design the details on the administration of HD.

  1. Lines 242-244: “The percentage area of immunoreactivity (determined by the number of positive pixels) was expressed as % of total tissue area (red staining).” What was the threshold for the IHC signal to be considered “positive”? From the description in Methods it is unclear what the red color represents and thus how this staining can be used for the normalization of IHC signal. Also, why not to simply normalize by the area of the AOI?

As suggested by the referee, we rewrote the part of the immunohistochemistry analysis in the Methods section (Lines 190-198).

  1. Line 316: “The number of neurons in the SN of rotenone mice was reduced compared with control mice (Fig. 2B, D)”. Fig 2D shows not the number of neurons (which should indeed be shown on an additional panel), but cell injury score.

Thank you for your suggestion, we rewrote this sentence (Lines 285-291).

  1. Fig 3: Please, proofread the legend as some references to the panels don’t correspond to the data that’s shown. Fonts on panels D, H and I are too small. Panels A-C and E-G show redundant data, so panels E-G can either be removed or shown as a supplementary figure.

As suggested by the referee, we proofread the legend of Fig. 3 (now Fig. 2) and we have arranged the panels to correspond to the figures indicated. We also decided to show stereological data as a supplementary figure and enlarged the characters of the graphs.

  1. Figs 2-5. Scale bars are too small and very difficult to see.

Thank you for your suggestion, we inserted in the figure legends the scale bars.

  1. Fig 5. Please, specify what type of aSyn is recognized by the antibodies. Also, it is difficult to see whether aSyn staining co-localizes with TH on the merged images on panels E-G; these need to be shown as separate color channels.

As suggested by the referee, we have specified in Fig 5 (now Fig 2) which type of a-syn is recognized by our antibody. We also separated the color channels in order to better show the expression of TH and a-syn and their co-localization.

  1. Figs 10B and 11C appear to show redundant data. Also, please use the same style for bar graphs of the same group on all figures.

As suggested by the referee, we used the same style for bar graphs of the same group on all figure.

  1. Fig 13. There are no controls, quantification or any indication of the number of repeats of this experiment. These data either need to be finalized or removed from the current manuscript.

Thanks for your suggestion, we inserted the control and quantification in Fig. 13 (now Fig. 7). Moreover, we added in the figure legend the number of repeats of this experiment.

  1. In Methods (line 115), Hidrox is stated to contain 40-50% hydroxytyrosol, while in Discussion the number is 50-70% (line 570).

Thanks for your suggestion, we have corrected the typographical error

Reviewer 3 Report

See attached.

Author Response

The article deals with an interesting subject which is the potential neuroprotective effect of Hidrox® in an animal model of Parkinson’s disease. The study provides remarkable evidences on the ability of this extract to activate the adaptive stress pathway in the potential treatment of degenerative diseases. 

In my opinion, the study is well designed in order to achieve the objectives and the methodology is correct.

However, Figures should be arranged to appropriately and clearly illustrate the findings.

Thank you for your suggestion, we rearranged the figures to appropriately illustrate the results.

In Material and Methods section, the description of distribution of samples from animals for histological and Western blot techniques must be clarified. It is not clear if a total of 10 animals for each technique are used or samples from the same 10 animals are distributed for both of them.

Thank you for your suggestion, we inserted in the figure legends the number of the animals that we used for the analysis.

In histology, semi-quantitative rating is defined, indicating ‘Representative images are shown’. But those images are not included in the Manuscript. They should be. Moreover, in experimental group, Group 2 (Sham + vehicle) is described, but data are not shown. This should be justified. Or maybe, the group 2 should be excluded from this study…

Thank you for your suggestion, we rewrote the sentence because we intended that images are representative of all animals in every group (Lines 169-171). Moreover, we included in the histological results the Sham+HD group to demonstrate that there are no differences with the Sham group (Lines 285-291).

Concerning Results section, the Authors have used different symbols (* and #) in order to distinguish comparisons between groups, but as they consist of statistical symbols, the universal nomenclature should be appropriately used in all of them. 

As suggested by the referee, we used the same style for bar graphs and symbol of the same group on all figure

In Lines 314 and 315, ‘cellular inflammation’ and ‘invisible nuclei’ are not correct terms. They must be substituted by the pathological technical terminology.  

As suggested by the referee, we rewrote the sentence using the correct pathological technical terminology (Line 288).

All immunohistochemical images (not confocal) should be edited by experts in order to illustrate the findings with the real colours. Furthermore, groups from which each sample comes and marker used must be clearly indicated in the panel. It is really hard to understand the results provided by means of the Figures at present configuration.   

As suggested by the referee, we had modified the immunohistochemistry images by experts in order to illustrate the results with real colors. We also wrote the names of the various groups and the markers under investigation in the panels.

In Figures 4 and 13 only two groups are compared while three are in the rest of Figures.

Thank you for your suggestion, we wrote better the figure legends in Fig 4 (now Fig. 2) and in Fig. 13 (now Fig. 7).

In Figure 5, Lines 391 and 392 are referring to E – G instead of A – C. 

Thank you for your suggestion, we corrected the typographical error in Fig 5 (now Fig 2 and 3).

Discussion section is appropriate. Only in Conclusions, the last sentence should be re-written in order to reflect the real impact of the findings. It sounds too much ambitious and it could even raise excessive and unreal expectations. 

As suggested by the referee, we rewrote the conclusions.

Some minor points:

  • In Material and Methods section, commercial reference details are missing for some reagents used.

Thanks for your suggestion, we have reviewed the Material and Methods section and added the commercial reference details to all reagents (Lines 251-252).

  • In Figure S1, the abbreviation GC-MS must be defined.

Thanks for your suggestion, we have defined the abbreviation GC-MS (Line 133)

  • Some spaces and words appear duplicated (for instance, line 128…..of of HD….)

Thank you for your suggestion, we have corrected all typographical errors.

Reviewer 4 Report

The paper by Siracusa et al investigates neuroprotective efficacy of Hidrox (HD) in a mouse model of rotenone-induced toxicity. Although overall the study is well designed and the data support the hypothesis that hydroxytyrosol can protect against oxidative stress in vivo, several questions need to be addressed before the manuscript is ready for publication.

Major:

  1. Methods (Line 136): “Group 2: Sham + HD = HD solution was administrated by i.p. for 28 days (data not shown; N=10)”.  Why is this group not shown? For some of the presented data, such as Figs 10-12, it is critical to investigate whether HD alone has the same effect as HD+Rotenon.
  2. Individual data points need to be shown on bar graphs on all figures. Was the number of samples always 10/group on all the figures? If not, indicate N in figure legends.

Other:

  1. Methods (Line 140): Please, provide more details about drugs administration. Was HD injected before rotenone? Was there a delay between the second drug delivery?
  2. Lines 242-244: “The percentage area of immunoreactivity (determined by the number of positive pixels) was expressed as % of total tissue area (red staining).” What was the threshold for the IHC signal to be considered “positive”? From the description in Methods it is unclear what the red color represents and thus how this staining can be used for the normalization of IHC signal. Also, why not to simply normalize by the area of the AOI?
  3. Line 316: “The number of neurons in the SN of rotenone mice was reduced compared with control mice (Fig. 2B, D)”. Fig 2D shows not the number of neurons (which should indeed be shown on an additional panel), but cell injury score.
  4. Fig 3: Please, proofread the legend as some references to the panels don’t correspond to the data that’s shown. Fonts on panels D, H and I are too small. Panels A-C and E-G show redundant data, so panels E-G can either be removed or shown as a supplementary figure.
  5. Figs 2-5. Scale bars are too small and very difficult to see.
  6. Fig 5. Please, specify what type of aSyn is recognized by the antibodies. Also, it is difficult to see whether aSyn staining co-localizes with TH on the merged images on panels E-G; these need to be shown as separate color channels.
  7. Figs 10B and 11C appear to show redundant data. Also, please use the same style for bar graphs of the same group on all figures.
  8. Fig 13. There are no controls, quantification or any indication of the number of repeats of this experiment. These data either need to be finalized or removed from the current manuscript.
  9. In Methods (line 115), Hidrox is stated to contain 40-50% hydroxytyrosol, while in Discussion the number is 50-70% (line 570).

Author Response

This study aims to determine the in vivo effects of hydroxytyrosol, in the form of Hidrox (HD), on the neurobiological and behavioral alterations induced during rotenone administration to wildtype male mice. Results indicate that HD may have the potential to reduce the loss of dopaminergic neurons and damage associated with this loss. Overall, this is a straightforward study with promising results that requires some streamlining for readability and a more conservative interpretation of results. Throughout the manuscript, it needs to be very clear that the HT product was provided at the same time as the rotenone-induction occurred (i.e. it was not given as a treatmentbut rather as a “preventative”).

Major Revisions:

  • A hypothesis for this study was never explicitly stated and should be included.

Thank you for your suggestion, we have included a hypothesis for our study in the introduction and discussion section.

  • Both the introduction and discussion sections require rewriting, with each paragraph providing a certain piece of essential information relevant to the study. For example, an entire section is provided in the introduction on Nrf2 but no part of this pathway is actually assessed in this experimental model. Why wasnt Nrf2 or its targets included?

As suggested by the referee, we rewrote the introduction and discussion section. In addition, we added the data on Nrf2 as we hypothesize that HD acting through this pathway regulates oxidative stress and neuroinflammation underlying Parkinson's disease.

  • In the discussion, the first paragraph should include a statement about the overall findings in this study. Then, each paragraph should not repeat results but integrate these findings into the overall conclusion. A figure summarizing how the protein markers of apoptosis, (Fig 6), NF-Kb (Fig 7), inflammasone (Fig. 8, 9), heat shock proteins (Fig 10 and 11) and γ-GCS (Fig. 12) culminate in Parkinsons pathology would be useful.

As suggested by the referee, we rewrote the discussion and inserted a figure summarizing the pathways we examined and which are implicated in Parkinson's disease.

  • Why were drug concentrations of HT or its metabolite not assayed in neural tissue? The study describing the pharmacokinetics of hydroxytyrosol used IV injection to demonstrate tissue distribution, not IP. Additionally, how was the IP dose of HT selected?

In previous Parkinson's studies we have already used the intraperitoneal administration (Siracusa, R. et al. 2015; Paterniti, I. et al. 2017; Siracusa, R. et al. 2017). With regard to the chosen dose, we referred to the following studies:

  • Terzuoli et al. 2010 and 2016 showed that HT (10 mg / kg i.p.) attenuates the growth of colon cancer cells by modulating the inflammatory response.
  • Shangha et al. 2013 showed that HT (10 mg/kg i.p.) protects the liver from oxidative damage following ischemia / reperfusion.
  • Carito et al. 2015 showed that 10 consecutive days of a mixture of polyphenols (10 mg/kg i.p.) containing mainly hydroxytyrosol extracted from olive pomace and dissolved in saline solution reduced cytokine levels in an experimental model of paw edema.

Moreover, we have not measured the pharmacological concentrations of HT or its metabolite in neuronal tissue because studies already exist (María-Carmen López de las Hazas et al. 2018).

  • A more thorough description of the pathologic classification is required. What exactly was included in mild, moderate, severe and more severe categories? A supplemental table could be provided.

We want to clarify that we used a modified version of the histological score made by Kawai and Akira 2007. In particular we counted injured neurons following these criteria:  0=normal, no death neuron observed; 1=insignificant pathology, SN contained one to five death neurons; 2=modest pathology, SN contained five to 10 death neurons; 3=severe pathology, SN contained more than 10 death neurons; 4=more severe pathology, SN contained only death neurons.

  • Figures 6-11 represent Western blot analyses for various proteins. It is recommended that the authors either (1) reduce the overall number of figures to better integrate and describe these findings or (2) place some of the data in supplemental. As the results section currently reads, these proteins appear “cherry-picked” and all demonstrate similar results. Were any proteins investigated that demonstrated no difference between experimental groups?

As suggested by the referee, we reduced the number of figures in order to make it easier to read the results. Moreover, we only investigated the proteins linked to the Nrf2 pathway and that HD could modulate.

  • Figure 13 – where is the sham group?

As suggested by the referee, we inserted the sham group in Fig. 13 (now Fig. 7)

  • Study limitations are required in the discussion. There was no mention of sex effects (only male mice were used) and the fact that the HD was given at the same time as rotenone induction limits conclusions to preventative, but not therapeutic, applications.

As suggested by the referee, we rewrote the discussion and inserted the hypothesis and the limitations of our study (Lines 571-576)

Minor Revisions:

  • Abstract: Line 33-34 is not a complete sentence

As suggested by the referee, we rewrote the sentence (Lines 32-35)

  • Abstract: The conclusion needs to be rewritten. This study did not demonstrate the safety

profile and therefore should not be included as a conclusion in the abstract.

            As suggested by the referee, we rewrote the conclusion of the abstract (Lines 38-39)

  • Introduction, Line 61, reword scientifically proven

As suggested by the referee, we rewrote the whole introduction.

  • Introduction: Line 62: This reference describes hydroxytyrosols ability to cross the bloodbrain

barrier but specifically indicates that brain update is lower compared to other organs.

We rewrote the introduction and removed this sentence.

  • Introduction: Line 74: However an HT treatment” – please describe that this was given at

the same time as the rotenone-induction model was initiated.

We rewrote the introduction

  • Introduction: Line 93, “…olive pulp containing 50-70% HT in an animal model.

Thanks for your suggestion, we have corrected the typographical error

  • Methods- please provide reasoning for the dose and duration of oral rotenone administration. While the researchers clearly demonstrate that this dose induces disease, other studies have used 30 mg/kg for up to 56d (Inden M et al. 2011). Also, it needs to be included in results that the sham + HD group demonstrated no differences from the Sham group as this data is not shown.

We decided to use the daily dose of 5 mg / kg for 28 days based on the study by Yan Liu et al. 2015 in which they demonstrate that this dose has a low mortality and also at 4 weeks it is capable of inducing behavioral changes, loss of dopamine and dopaminergic neurons and accumulation of a-synuclein.

Moreover, as suggested by the referee, we included in the histological results the Sham+HD group to demonstrate that there are no differences with the Sham group (Lines 285-291).

  • Results Line 286 provide a reference for the pole test as a suitable test for bradykinesia.

As suggested by the referee, we inserted a reference for the pole test as a suitable test for bradykinesia (Line 274).

  • For Figure 1, data at time point 0 is not shown and it is not mentioned that no significant differences were observed. For all data results depicting mouse quantitative data (i.e. all bar graphs), box and whisker plots are preferred, as a more accurate representation of the biologic variability is apparent. From Fig. 3 onward, figure legends are difficult to follow and should be rewritten (i.e. A should always be described first, Bis missing from Fig 3). Fig. 5 E-G need a scale bar.

As suggested by the referee, we inserted in Results section a sentence to justify why we don't show the data at time 0 (Lines 270-273). Moreover, we changed the graphs, rewrote the figure legends and inserted the scale bar in the legend of figure 5 (now Fig 2 and 3)

  • Line 328- reword, In the SN after rotenone administration was demonstrated an important…”

As suggested by the referee, we rewrote the sentence (Lines 313-314)

  • Discussion line 550: Information on the benefits of the Mediterranean diet would be useful in the introduction as well. A reference is needed for line 554 Highest antioxidant

As suggested by the referee, we wrote a part on the benefits of Mediterranean diet in the discussion (Lines 502-510) and inserted the requested reference (Line 508).  

  • Discussion line 590: Provide references for the use of these behavioral tests for this

phenotype.

We rewrote the discussion and eliminated this part.

  • Discussion line 605-606: Reword As we said earlier

We rewrote the discussion

  • Discussion line 630: Reword slow down

We rewrote the discussion

  • Discussion lines 684 and 687-688: This study does not show therapeutic value, only

            preventative. This needs to be clearly stated.

We rewrote this part of the discussion.

Round 2

Reviewer 2 Report

Despite the Authors have followed some suggestions, some others remain uncorrected. 

-  The Authors have used different symbols (* and #) in order to distinguish comparisons between groups, but as they consist of statistical symbols, the universal nomenclature should be appropriately used in all of them.

- The Authors inserted in the figure legends the number of the animals that we used for the analysis. However, it must be clearly indicated in Material and Methods section.

- Figures should be arranged to appropriately and clearly illustrate the findings. All immunohistochemical images should be edited by experts in immunohistochemical images in order to illustrate the findings with the real colours.

- Commercial reference remains undetailed for some reagents used in Material and Methods section.

Author Response

The Authors have used different symbols (* and #) in order to distinguish comparisons between groups, but as they consist of statistical symbols, the universal nomenclature should be appropriately used in all of them.

- Thank you for your suggestion, we used the universal nomenclature in the figures and figure legends.

The Authors inserted in the figure legends the number of the animals that we used for the analysis. However, it must be clearly indicated in Material and Methods section.

- As suggested by the referee we inserted the number of the animals also in the Material and Methods section.

Figures should be arranged to appropriately and clearly illustrate the findings. All immunohistochemical images should be edited by experts in immunohistochemical images in order to illustrate the findings with the real colours.

- As suggested by the referee, we rearranged the figures and the immunohistochemical images were edited by experts.

Commercial reference remains undetailed for some reagents used in Material and Methods section.

-Thank you for your suggestion, we inserted the details for all reagents used in Material and Methods section.

Reviewer 3 Report

Thank you for the revisions. The results now tie in nicely to the hypothesis and provide evidence for the role of Nrf2.

My only remaining required edit is to define the limitations to this study (i.e. HT was given AFTER induction of disease and sex effects were not evaluated) in the discussion. Sex effects were only simply mentioned in the conclusion as future studies and I did find any included discussion about timing of HT administration.

Other than this, only minor edits are required at this stage.

Introduction: Spell out “HT” for the first time (line 71).

Line 101, “we hypothesized”

Line 514-515, reword “thanks to a unique manufacturing process”

Line 526: reword, “thanks to our PD model…”

Line 540, reword, “…our findings supported our hypothesis that HD carries out…”

Line 542, reword, “As an effect of…”

Line 568, “sin-synuclein”? Should be alpha-synuclein

Author Response

Thank you for the revisions. The results now tie in nicely to the hypothesis and provide evidence for the role of Nrf2.

My only remaining required edit is to define the limitations to this study (i.e. HT was given AFTER induction of disease and sex effects were not evaluated) in the discussion. Sex effects were only simply mentioned in the conclusion as future studies and I did find any included discussion about timing of HT administration.

- Thanks for your suggestion, we wrote also in the discussion the limitations of our study.

Other than this, only minor edits are required at this stage.

Introduction: Spell out “HT” for the first time (line 71).

Line 101, “we hypothesized”

Line 514-515, reword “thanks to a unique manufacturing process”

Line 526: reword, “thanks to our PD model…”

Line 540, reword, “…our findings supported our hypothesis that HD carries out…”

Line 542, reword, “As an effect of…”

Line 568, “sin-synuclein”? Should be alpha-synuclein

- As suggested by referee, we rewrote the required sentences.

Reviewer 4 Report

Major:

As I was saying in the previous revision – knowing whether HD by itself is able to induce an increase in protein levels on Figs 4-6 is very important as this will predict a likelihood of side-effects if the drug is used as therapy. Is the effect of HD on the levels of Nrf2, HO1, Hsp70, etc additive to rotenone, or it has this effect only in the presence of rotenone (i.e. amplifies stress-induced changes)? An HD control without rotenone needs to be present on Figs 4-6. This is not so important for other figures where HD is shown to prevent deleterious changes induced by rotenone.

The authors did changed bar graphs to scatter plots on Fig 1, but this needs to be done for all figures. Scatter plots or bar and whiskers plots (with individual data point shown) are both appropriate.

Minor:

Fig 3 legend. First, there was no analysis of co-localization performed, so there can be no conclusion drawn about increased or decreased co-localization. Second, co-localization depends on the detection threshold and thus on the intensity of aSyn staining, so visually less co-localization in rotenone-treated slices might be due to increased overall aSyn fluorescence. I recommend avoiding using the term co-localization.

Fig 4, 8 and 9. It took me a while to understand that A1, C1, E1, D1 show quantification of the bands A, C, … This works on Figs 5 and 6, but not here. Simply add titles (i.e. Nrf-2, HO-1, etc) to each of the bar graphs and use regular A-E panel lettering. Was the tissue collected from a specific brain area or from the whole brain?

Figs 3-9. Please, clarify the statement “The data are representative of at least three independent experiments and are expressed as mean ± SEM from N = 5 mice/group.” Was it 3 experiments with 5 mice/group in each of them? The gels show 3 bands/group, so where does 5/group come from?

I am asking again to use the same style for bar graphs (or symbols for scatter plots) of the same group on all figures. For example, currently rotenone group is shown as a bar with: Fig 1 – horizontal lines; Fig 2 – crossed lines, Fig S2 – horizontal lines, Fig 4 – crossed lines, Fig 5, 6 – horizontal lines, Fig 7 – different type of horizontal lines. I also don’t believe the use of coloreds is prohibited by the journal.

WB group on Fig 7 is not mentioned anywhere else in the text. Also, please, add a description of how the gel was quantified. Was the intensity of the whole lane measured or of a particular band?

Line 516: The phrase “HD have shown a beneficial effect on the cardiovascular system, bacteria, cellular aging, nerves, UVB damaged skin and psoriasis” is very misleading as there is a difference between bacteria or psoriasis and the treatments of thereof.

Grammar:

A multitude of grammar mistakes and misuse of words, such as “nominated” instead of “named”, “value” instead of “evaluate”, “furtherly” instead of “further”, etc.

Lines 167-169. “death neurons” should be changed to “dead neurons”. Also, what was counted as a dead cell with H&E staining? Chromatin condensation? Then it's not dead yet, but undergoes necrosis or apoptosis.

Line 342. This is not a complete sentence.

Line 345 should read “Animals treated with rotenone and HD showed increased staining for DAT (G) compared with rotenone group.”

Line 389. should be GCS instead of GGCS

Hidrox is misspelled on Fig 5 and 6.

NF-κB is misspelled on Fig 8.

Sentence on Lines 546-547 is incomplete.

Line 568. should be alpha-synuclein instead of sin-synuclein. Also, the link between aSyn accumulation and motor deficits is only a hypothesis.

Author Response

As I was saying in the previous revision – knowing whether HD by itself is able to induce an increase in protein levels on Figs 4-6 is very important as this will predict a likelihood of side-effects if the drug is used as therapy. Is the effect of HD on the levels of Nrf2, HO1, Hsp70, etc additive to rotenone, or it has this effect only in the presence of rotenone (i.e. amplifies stress-induced changes)? An HD control without rotenone needs to be present on Figs 4-6. This is not so important for other figures where HD is shown to prevent deleterious changes induced by rotenone.

- We thank the referee for the comments, we apologize for not showing the effects of HD alone in figures 4-6. we decided not to investigate more the HD group alone because from our preliminary results, shown in the first review in the histological examination, and in accordance with the data in the literature, HT alone does not cause changes in protein levels compared to the control group  (https://doi.org/10.1016/j.biopha.2018.11.120). now it would be impossible for us to repeat the experiment due to the COVID-19 emergency. however, we realize the importance of consolidate the result from a therapeutic point of view avoiding possible side effects, and that is why we will take the referee's suggestion into strong consideration for our subsequent studies.

The authors did changed bar graphs to scatter plots on Fig 1, but this needs to be done for all figures. Scatter plots or bar and whiskers plots (with individual data point shown) are both appropriate.

- As suggested by the referee, we changed the type of graph in all our figures.

Minor:

Fig 3 legend. First, there was no analysis of co-localization performed, so there can be no conclusion drawn about increased or decreased co-localization. Second, co-localization depends on the detection threshold and thus on the intensity of aSyn staining, so visually less co-localization in rotenone-treated slices might be due to increased overall aSyn fluorescence. I recommend avoiding using the term co-localization.

- As suggested by the referee we have avoided using the term co-localization in the Fig 3 legend.

Fig 4, 8 and 9. It took me a while to understand that A1, C1, E1, D1 show quantification of the bands A, C, … This works on Figs 5 and 6, but not here. Simply add titles (i.e. Nrf-2, HO-1, etc) to each of the bar graphs and use regular A-E panel lettering. Was the tissue collected from a specific brain area or from the whole brain?

- As suggested by the referee we have indicated the names of the antibodies investigated in the graphs corresponding to bands A, C etc. in Figs 4, 8 and 9. For western blot analysis observed in these figures we used the whole brain.

Figs 3-9. Please, clarify the statement “The data are representative of at least three independent experiments and are expressed as mean ± SEM from N = 5 mice/group.” Was it 3 experiments with 5 mice/group in each of them? The gels show 3 bands/group, so where does 5/group come from?

- For the in vivo studies, n represents the number of animals studied. In the experiments involving histology or immunohistochemistry, the figures shown are representative of at least three experiments (histologic or immunohistochemistry coloration) performed on different experimental days on the tissue sections collected from 5 animals in each group. The western blots analyses are representative of 3 different gels made by dividing the number of samples obtained from 5 animals for each experimental group in different days.

I am asking again to use the same style for bar graphs (or symbols for scatter plots) of the same group on all figures. For example, currently rotenone group is shown as a bar with: Fig 1 – horizontal lines; Fig 2 – crossed lines, Fig S2 – horizontal lines, Fig 4 – crossed lines, Fig 5, 6 – horizontal lines, Fig 7 – different type of horizontal lines. I also don’t believe the use of coloreds is prohibited by the journal.

- As requested by the referee we have used the same style for the graphs of all figures

WB group on Fig 7 is not mentioned anywhere else in the text. Also, please, add a description of how the gel was quantified. Was the intensity of the whole lane measured or of a particular band?

- The bands were quantified normalizing pixels in each lane to the loading control band.  This sentence has been added in the text.

Line 516: The phrase “HD have shown a beneficial effect on the cardiovascular system, bacteria, cellular aging, nerves, UVB damaged skin and psoriasis” is very misleading as there is a difference between bacteria or psoriasis and the treatments of thereof.

- Thank you for your comment, we removed this sentence to avoid misunderstandings.

Round 3

Reviewer 4 Report

I have no further concerns with this manuscript.